# Surface Photovoltage Response of ZnO to Phosphate-Buffered Saline Solution with and without Presence of *Staphylococcus aureus*

**DOI:** 10.3390/nano13101652

**Published:** 2023-05-16

**Authors:** Dustin A. Johnson, John M. Reeks, Alexander J. Caron, Shauna M. McGillivray, Rafal J. Wiglusz, Yuri M. Strzhemechny

**Affiliations:** 1Department of Physics & Astronomy, Texas Christian University, Fort Worth, TX 76129, USA; y.strzhemechny@tcu.edu; 2Institute of Low Temperature and Structure Research, Polish Academy of Sciences, 50-422 Wroclaw, Poland; j.reeks@intibs.pl (J.M.R.); r.wiglusz@intibs.pl (R.J.W.); 3Department of Biology, Texas Christian University, Fort Worth, TX 76129, USA; a.j.caron@tcu.edu (A.J.C.); s.mcgillivray@tcu.edu (S.M.M.)

**Keywords:** ZnO, antibacterial, *S. aureus*, phosphates, antimicrobial, surface photovoltage

## Abstract

Nano- and microscale zinc oxide (ZnO) exhibits significant potential as a novel antibacterial agent in biomedical applications. However, the uncertainty regarding the underlying mechanisms of the observed antimicrobial action inhibits the realization of this potential. Particularly, the nature of interactions at the free crystalline surface and the influence of the local bacterial environment remains unclear. In this investigation, we utilize ZnO particles synthesized via tunable hydrothermal growth method as a platform to elucidate the effects of interactions with phosphate-rich environments and differentiate them from those with bacteria. This is achieved using the time- and energy-dependent surface photovoltage (SPV) to monitor modifications of the surface electronic structure and surface charge dynamics of the ZnO particles due to these interactions. It is found that there exists a dramatic change in the SPV transients after exposure to phosphate-rich environments. It also presents differences in the sub-bandgap surface electronic structure after these exposures. It can be suggested that these phenomena are a consequence of phosphate adsorption at surface traps corresponding to zinc deficiency defects. This effect is shown to be suppressed in the presence of *Staphylococcus aureus* bacteria. Our results support the previously proposed model of the competitive nature of interactions between *S. aureus* and aqueous phosphates with the free surface of ZnO and bring greater clarity to the effects of phosphate-rich environments on bacterial growth inhibition of ZnO.

## 1. Introduction

Wurtzite zinc oxide (ZnO) is an abundant type II–VI semiconductor with a wide, direct band gap of ~3.37 eV at room temperature, which corresponds to the spacing between the vacant 4*s* orbitals of Zn^2+^ and the occupied 2*p* states of O^2−^ atoms within the crystalline lattice. Notable physical and chemical stability [1], in addition to a range of useful optoelectronic properties [1], has resulted in extensive and diverse applications in a variety of fields. At present, budding applications center around its antimicrobial properties which are both potent and well documented [2,3,4]. Such antimicrobial applications are of critical interest due to the increasing threat to global health and food security posed by bacterial infections, particularly those antibiotic-resistant strains which are rapidly becoming both more prevalent and difficult to treat. Powerful bacterial adaptive responses in the form of genetic mutations and lateral gene transfer make the development of effective traditional antibiotics increasingly difficult and less profitable, thus necessitating the search for nontraditional alternatives for usage in sanitation and medicines [5,6].

Nano- and microscale ZnO is an attractive candidate in this regard, as it exhibits reduced toxicity to human cells, is recognized as a safe substance for exterior applications by the US Food and Drug Administration [7,8], and demonstrates increased selectivity towards bacteria with efficacy against existing antibiotic-resistant strains [9,10]. ZnO compounds are seen to exhibit effectiveness against a wide range of both Gram-positive and Gram-negative bacteria under various conditions with selective toxicity [11,12] and effectiveness in combating biofilm formation. In addition to the inherent antimicrobial characteristics, ZnO is an inexpensive, readily available material with significant photocatalytic efficiency and synthesis methods that are both simple and scalable. The properties have led to a proposed usage as an antibacterial agent in key industries such as healthcare, water treatment, textiles, food storage, biomedicine, and transportation [3,4,13,14]. The pursuit and development of efficient, novel bactericidal applications, however, is inhibited by uncertainty surrounding the underlying mechanisms of the observed antimicrobial action. There exists a debate surrounding the most suggested mechanisms, namely, the generation of various reactive oxygen species (ROS), the release of Zn ions, and surface–surface interactions between the bacterial cell wall and the free crystalline surface of ZnO [15,16]. The relative dominance of these mechanisms is also subject to debate, thus inhibiting the efficient design of novel ZnO-based antimicrobials. A definitive description of the fundamental mechanisms underlying these behaviors is therefore desirable from the perspectives of both scientific interest and practical application.

To this end, significant progress has been made of late, with researchers taking a biological approach and investigating the bacterial response to various ZnO compounds in varying environments. These works emphasize the role of both the free crystalline surface and the ambient environment in which these antibacterial interactions are to take place. Studies such as those by Xu et al. [17] and Liu et al. [18] have highlighted the role of defects in ROS generation, although their role in other proposed mechanisms is still unclear. Numerous studies, such as those performed by Zakharova et al. [19], demonstrate a dependence of bacterial growth inhibition in ZnO on the bacterial growth media, yet descriptions of the fundamental interactions involved are lacking. Our previous work has established ZnO microparticles (MPs) as a useful platform for studying the antimicrobial mechanisms at the nanoscale and they were utilized to outline the general effects of exposure to phosphate-rich environments on the defect structure of ZnO and differentiate those with bacteria vs. without bacteria [20]. Within this context, we propose that surface modification by media of the free crystalline surface of ZnO can inhibit or enhance the antibacterial efficacy and, specifically, that adsorption of aqueous phosphate species onto surface defects can inhibit antimicrobial behavior by restricting direct interaction of these sites with bacteria. To support such a hypothesis, one must confirm the surface specificity of the interactions in question and provide evidence for specific sites of this adsorption in comparison to those attributed to antibacterial interactions. Addressing this, we present both time- and energy-dependent surface photovoltage (SPV) studies of ZnO particles before and after exposure to *Staphylococcus aureus* (*S. aureus*) bacteria in the phosphate-buffered saline (PBS) solution, as well as exposure to just PBS alone. Phosphate compounds are abundant in biological environments, and due to their strong metal-complexation capability, phosphate-induced transformations play an important role in the behaviors and toxicity of ZnO-based nanomaterials [21]. SPV is very sensitive to changes in the surface potential associated with the surface charge dynamics, electronic structure, and occupation of surface defects by impurities [22]. Such experiments are of interest as they allow for the elucidation of the surface-specific responses of ZnO to bacteria and the local environment. The usage of ZnO MPs serves to control for internalization effects which are not thought to be a driving force in the observed bacterial growth inhibition [23]. Exposure of ZnO MPs to PBS in isolation allows for differentiation of interactions at the surface between bacteria and the media. These interactions can overlap in nature, and, thus, such control is vital for the accurate interpretation of results.

In this work, we find that there exists a dramatic change in the rate and magnitudes of the processes contributing to the contact potential difference (ΔCPD) after exposure to phosphate-rich environments, particularly for longer-lived processes that may be heavily influenced by adsorbed species at the surface. We also note that the presence of bacteria has an impact on faster SPV processes and suppresses the effect of PBS media at longer timescales. In addition, the SPV spectra demonstrate changes in the electronic structure because of exposure to PBS, which indicates significant adsorption of phosphate compounds at oxygen-rich sites on the crystalline surface. Such results bring greater clarity to the nature of phosphate-rich media interactions with ZnO and point to a restriction of specific surface defects that may be influential for the antimicrobial mechanisms of ZnO.

## 2. Materials and Methods

### 2.1. ZnO MP Synthesis and Morphological Characterization

The ZnO MPs utilized here were synthesized via a bottom-up hydrothermal growth method. An equimolar solution of 1 M zinc acetate dihydrate [Zn(CH_3_CO_2_)_2_·H_2_O] and 1 M hexamethylenetetramine [(CH_2_)_6_N_4_] (HMTA) was produced in deionized (DI) water. The HMTA was added to DI water and left to stir continuously for 5 min to begin the dissociation of the water molecules. After this, zinc acetate dihydrate was added to the HMTA solution and mixed for 30 min to form ZnO precursors. An additional 1 cm^2^ strip of 99.999% pure Zn foil supplied by Sigma Aldrich (St. Louis, MO, USA) was then added after mixing to serve as an additional source of free Zn. This solution was transferred into a Teflon container and sealed in a stainless-steel autoclave before undergoing heat treatment at 100 °C for 3 h in a Stabletemp forced air drying oven (Across International, Sparks, NV, USA). Following this baking period, ZnO was formed as a precipitate, so the solution went through centrifugation and subsequent removal of the organic supernatant. The solid material then was rinsed via 7 cycles of alternating DI water and acetone. 

The morphology of the resulting ZnO MPs was determined via surface area calculations performed with ImageJ software (version 1.51) on images of the particles’ surfaces captured with scanning electron microscopy (SEM) utilizing a JEOL FE-SEM instrument (JEOL, Peabody, MA, USA) at an operating voltage of 15.0 kV. The ZnO powder was pelletized and attached to an aluminum SEM mount with carbon tape prior to imaging.

### 2.2. Biological Exposure

ZnO MPs were exposed to PBS and PBS containing *S. aureus* by performing minimum inhibitory concentration (MIC) assays. The details of these studies were described in depth previously [20]. In short, the PBS utilized in these studies was prepared through the dissolution of two salts: sodium chloride (NaCl) (4 g) and potassium chloride (KCl) (0.1 g) in 500 mL of water. An additional 0.72 g of anhydrous sodium phosphate–dibasic was dissolved as a phosphate source and the pH was adjusted to 7.4. The solution was then autoclaved prior to the addition of bacteria and usage in biological assays. A methicillin-susceptible *S. aureus* strain in the Newman strain was utilized in our studies. Bacteria were grown to an early log phase with an optical density at 600 nm excitation (OD600) of 0.4 in MHB. The OD600 measurements were performed in a 96-well plate using a Fluostar Omega plate reader (BMG Labtech, Cary, North Carolina, USA). They were then washed before suspension in PBS. These cultures were subsequently diluted with ZnO in PBS and inverted at 37 °C. 

### 2.3. Surface Photovoltage Studies

In vacuo surface photovoltage (SPV) characterization was performed on both as-grown ZnO MPs and those same particles following exposure to *S. aureus* bacteria in PBS as well as PBS without bacteria. SPV is a nondestructive and highly surface-sensitive probing technique in which one monitors the surface potential due to changes in illumination conditions: incident wavelength (SPV spectroscopy) or time (transient SPV). A Besocke Delta Phi GmbH (Jülich, Germany) Kelvin Probe S was positioned near the surface of a sample forming a parallel plate arrangement between the reference electrode and the sample. This configuration allows for indirect surface potential measurement via its relationship with the surface work function. In this arrangement, a contact potential difference (ΔCPD) is generated between the sample surface and the probe resulting in an electric field arising at the junction. This field is nullified by an external DC bias that is proportional to the difference in the work functions of the materials. SPV spectral response was monitored with respect to light supplied by a fiber optic bundle coupled through an optical feedthrough with an ex vacuo optical train consisting of a 250 W QTH lamp as a white light source, a pair of fused silica lenses, bandpass filters, and the Oriel Cornerstone grating monochromator (Newport, Rochester, NY, USA). The surface and near-surface electronic states are highly sensitive to the environment and the presence of adsorbed species. For this reason, in our studies, the SPV experiments were performed in a high-vacuum chamber under 10^−8^ Torr. In addition to the SPV spectra, transient response curves were obtained before each run. The samples were illuminated directly with white light until the SPV saturation was achieved and subsequently quenched in the dark until surface state equilibrium was achieved.

## 3. Results and Discussion

The particles used in our studies have dimensions of ca. 1 micron with a well-defined hexagonal prism structure, as determined by the FE-SEM probe (see Figure 1). These crystals present a useful platform for our investigations due to the presence of different surface types. Alternating planes of atoms along the *c*-axis make the hexagonal polar faces carry more excess charge and thus undergo significant surface reconstruction in comparison to the rectangular nonpolar faces [24,25]. This renders the hexagonal faces more defect-rich with complex electrochemical properties different from those of the rectangular surfaces [26,27].

A detailed description of the optoelectronic and physicochemical properties of these crystals as well as their contributions to the antibacterial action of ZnO can be found in previous work [20,23]. In this paper, we focus primarily on the surface-specific changes in the charge dynamics and electronic structure of ZnO particles due to their exposure to *S. aureus* bacteria in the PBS solution as well as PBS without the presence of bacteria. 

Surface charge dynamics of these crystals were investigated by SPV transients utilizing panchromatic light for the sample depicted in Figure 1, first as-grown and then following exposure to PBS alone and PBS containing *S. aureus*. SPV transients in micro/nanoscale materials have complex origins due to several factors, including, but not limited to, a large active surface area, complex geometry, and modified width of the space charge region (SCR). Despite these challenges, the SPV signal could be comparable to that in bulk materials [28]. The results shown in Figure 2a demonstrate such complex transient behavior in our sample with multiple characteristic timescales of a negative contact potential difference (CPD) for all conditions observed (as to be expected in a compound semiconductor with an n-type conductivity [29,30]). Notable are the significant differences between the curves for different exposure conditions. The rate and the magnitude of changes in the CPD depend on the nature of the charge exchange between the surface states and the bulk, for which, as with an electrical network, the intensity and time of response to a step current can be approximated by a linear combination of exponential time dependencies, similar to those of a capacitor charging/discharging in a simple RC-circuit [31]:


(1)
$$\Delta V_{CPD}\left(t\right)=V_{0}+\overset{n}{\underset{i}{\sum }}V_{i}^{n}\left(1-e^{-\frac{t}{\tau_{i}}}\right)+\overset{m}{\underset{j}{\sum }}V_{j}^{m}e^{-\frac{t}{\tau_{j}}}$$


We used the model based on (Equation (1)) to fit the transient curves and thus quantify the observed changes. We applied the Levenburg–Marquardt fitting method [32] and determined that for all three experimental curves, Δ*V_CPD_*(*t*) is well fitted with four processes occurring on different timescales.

We compile the results of these fits in Table 1 and Figure 2b–d. It is worth noting that all the charge exchange processes were well fit with four decaying exponentials except for the “fastest” process in the sample exposed to PBS alone, which was fitted with a rising exponential. In comparing the parameters across the samples, it becomes apparent that the PBS medium significantly impacts the surface charge dynamics on all the timescales. The presence of *S. aureus* bacteria qualitatively alters the results for both *V_i_* and *τ_i_* compared to exposure to PBS alone. 

When considering the physical origins of these exposure-induced changes in the charge dynamics, there is a possibility of introducing new surface states and modifying existing ones. Especially prominent are the changes seen in the normalized *V*_4_ parameter (see Figure 3). Generally, in our fitting model, the *V_i_* parameters represent the effective charge “reservoir” of these processes analogous to the charge of a capacitive element.

The drastic increase of *V*_4_ and *τ*_4_ in addition to the change in the direction of the charge transfer for the fastest process after exposure to PBS indicates that the surface has drastically changed. The effects of PBS on increasing the characteristic time of the slowest process by an order of magnitude are well aligned with the results previously reported by us [20] pointing to the adsorption of phosphate groups at the surface of the ZnO MPs. The surface charge dynamics depend on the occupation of available energy states and their position with respect to the Fermi level. Therefore, transient processes occurring on different timescales could be produced by surface band bending resulting from the presence of adsorbed species. During illumination, holes are swept towards the bulk where they may recombine within surface states, thus reducing the band bending at the surface, allowing for electron tunneling and subsequent charge transfer between the crystalline bulk and adsorbed species at the surface. Such a process is relatively slow in comparison to the surface recombination of electrons with accumulated holes as there exists a surface barrier to overcome, accounting for this significant increase. The likelihood of such charge transfer occurring is highly dependent on the surface work function and the electronegativity of the adsorbed species. The abundance of oxygen atoms, as one of the most electronegative elements, makes such charge transfer probable given the adsorption of phosphate groups at the ZnO surface defect sites. Occupation of these surface states would have a significant impact through the distortion of the SCR, modification of existing surface states, and changing the surface dipole, all of which, in turn, would affect the surface charge transfer rates.

Adsorption of phosphate compounds is significant in the context of antibacterial action, as this phenomenon may limit access to surface sites potentially harmful to bacteria and stabilize the ZnO surfaces slowing their dissolution [21,33]. This adsorption is consistent with the previously reported [20] blue shift observed in the room-temperature luminescence spectra of the same ZnO MPs. The change in the direction of the surface charge flow of the fastest process could be attributed to the flipping of the occupancy of existing trap states due to the influence of the charges originally present in the molecules of the adsorbed layer.

Figure 4 shows characteristic times *τ_i_* for different conditions. One can observe that the presence of *S. aureus* mitigates the changes induced by PBS for slow processes, indicating that the presence of bacteria greatly limits or prevents the formation of the new traps introduced by PBS, consistent with the changes seen in Figure 3. The effects of bacteria on short timescales line up with reports of significant surface degradation of ZnO MPs in their presence [20]. By introducing extended defects at the free crystalline surface, direct bulk-to-trap recombination can occur, which is a very fast process compared to other possible surface charge recombination pathways. The effects on the intermediate timescales are less significant. We suggest that the observed exposure-induced changes are a result of modification of the existing states stemming from the partial dissolution of the crystal and adsorption of media components.

The mitigation by bacteria of the effects of PBS on slow processes (Figure 3 and Figure 4b,c) could be explained as follows. The presence of *S. aureus* may prevent or reverse the adsorption of phosphates at the ZnO surface. This effect can be attributed to either interaction of bacteria at potential phosphate adsorption sites or the removal of adsorbed phosphate groups themselves by bacteria. This, in turn, would indicate either overlapping interaction sites for aqueous phosphates and *S. aureus* bacteria or competition between ZnO MPs and media components for interaction with *S. aureus* bacteria. Further elucidation of possible mechanisms is provided by the SPV spectra. 

The results of SPV spectroscopy for the as-grown sample are shown in Figure 5 and summarized in Figure 6. In addition to a strong bandgap transition at ~3.3 eV, we observed the presence of several states in the bandgap: a ~1.5 eV trap-to-conduction band transition, a ~2.6 eV trap-to-conduction band transition, and a ~2.4 eV valence band-to-trap transition. Following theoretical calculations [34], we assign these transitions to oxygen vacancies, zinc vacancies, and oxygen interstitials, respectively. Assignment of these surface trap states is important as it allows us to define the initial nature of the free crystalline surface upon which interactions with bacteria and media components may take place. 

SPV spectra collected following the exposure to PBS with and without the presence of *S. aureus* bacteria demonstrated the effects on the surface electronic structure of ZnO. In addition to the anticipated weakening of the bandgap transition due to the partial dissolution of the surface and adsorption of media components, we find notable differences in the observed surface states, which we present in Figure 7. Transitions associated with the oxygen-rich states are suppressed following PBS exposure while the ~1.5 eV transition associated with the oxygen-deficient states is preserved. There have been several investigations into the kinetics of phosphate adsorption and it is widely attributed to the ligand exchange with surface hydroxyl groups resulting in PO_4_^3−^ adsorption [21,35]. Considering the highly reactive nature of an oxygen-rich surface in aqueous environments, it is clear that the formation of such hydroxyls and subsequent ligand exchange is a probable explanation for the removal of these states. Therefore, phosphate adsorption is likely to take place at these previously oxygenated sites, thus amplifying the slow charge-exchange processes and reversing the direction of the fast charge transfer discussed above.

Considering the results of Lv et al. in tracking Zn^2+^ ion release as a measure of ZnO particle solubility in aqueous phosphate environments [21], one can surmise that the high concentration of phosphates in a solution likely stabilizes the crystalline surface through the adsorption processes in addition to reacting with the initial rapid release of Zn^2+^ ions. This would have significant effects on the antibacterial action of these particles and supports previous studies linking phosphate-rich media with decreased bacterial growth inhibition [19,36]. Comparing these results with the SPV spectra obtained following the exposure to PBS containing *S. aureus* bacteria, we again observe a similar weakening of the bandgap transition due to the partial dissolution of ZnO crystals and adsorption of media components. However, we do not see significant changes to the sub-bandgap transitions observed in the sample exposed to PBS alone. In Figure 8 we see all three transitions that were originally present in the as-grown ZnO MPs. This lines up with the observed effects of bacteria containing PBS on the SPV transients discussed above, where we found little indication of new or suppressed states. The point of interest in these measurements lies in the apparent inhibition of the effects of PBS alone outlined thus far. The preservation of the oxygen-rich states is evidence of bacteria limiting phosphate adsorption at the free crystalline surface. This indicates that either bacteria significantly reduce the concentration of aqueous phosphates in the local environment or more likely that they promote increased solubility of the ZnO crystals through interactions at the crystalline surface, thus shifting the dominant phosphate interactions to reaction with Zn^2+^ in solution as opposed to surface adsorption. Such surface interactions are supported by the previously reported surface degradation and changes in the photoluminescence intensity after exposure to antibacterial interactions [20]. These findings suggest that bacteria promote Zn^2+^ ion release and this release is limited by aqueous phosphates, whereas interaction with the released ions is competitive in nature between bacteria and environmental phosphates. Bacteria and phosphates competing for interaction at the surface sites is unlikely due to the difference in their interactions at these states. Restriction of these sites by adsorbed species could limit surface reactive oxygen species (ROS) generation. Nevertheless, one would anticipate observation of a state that is suppressed by PBS but partially restored in the presence of *S. aureus* bacteria under the commonly proposed UV-mediated photocatalysis route of ROS generation in ZnO.

## 4. Conclusions

Nano- and microscale ZnO-based antibacterial agents represent viable and attractive candidates for novel applications. These include multifunctional dressings [37], nanostructured scaffolds [38], photocatalytic wearables [39], and more [40]. As such, insights into the fundamental interactions at the interface of the crystalline surface and bacteria cell wall as well as the solid–liquid interface between the free surface and the local bacterial environment are necessary to improve both the control and design of potential antibacterial agents. The findings presented here for the changes in the surface charge dynamics after exposure to PBS media with and without the presence of *S. aureus* demonstrate that there exist significant interactions at the free crystalline surface of ZnO. This surface is a highly dynamic and complex system with both oxygen-deficient and zinc-deficient defects despite the relatively high quality of the crystals utilized. The domination of slow processes in the SPV transients after exposure to PBS indicates that much of the surface interaction involves phosphate adsorption. The SPV spectra show the removal of oxygen-rich states after exposure to PBS, which we attribute to ligand exchange with hydroxyl groups at these oxygen-rich sites. We also provide evidence for the predicted competitive interactions between bacteria and environmental phosphate species through the preservation of the oxygen-rich defect states and suppression of the effects of PBS on slow surface charge exchange processes. The nature of these interactions suggests that the presence of *S. aureus* impacts particle solubility which would be influential in antimicrobial properties. Our results further elucidate the nature of phosphate interactions with ZnO by highlighting the occupation of zinc-deficient states at the surface that either limits their interaction with bacteria or stabilizes the surface, preventing further dissolution of the ZnO particles. The suppression of these effects in the presence of bacteria leads us to conclude that zinc-deficient surface sites are, likely, not relevant as direct interaction sites for bacteria although they do influence relevant interactions at the free crystalline surface of ZnO with both PBS and *S. aureus*, thus affecting bacterial growth inhibition.

## Figures and Tables

**Figure 1 nanomaterials-13-01652-f001:**
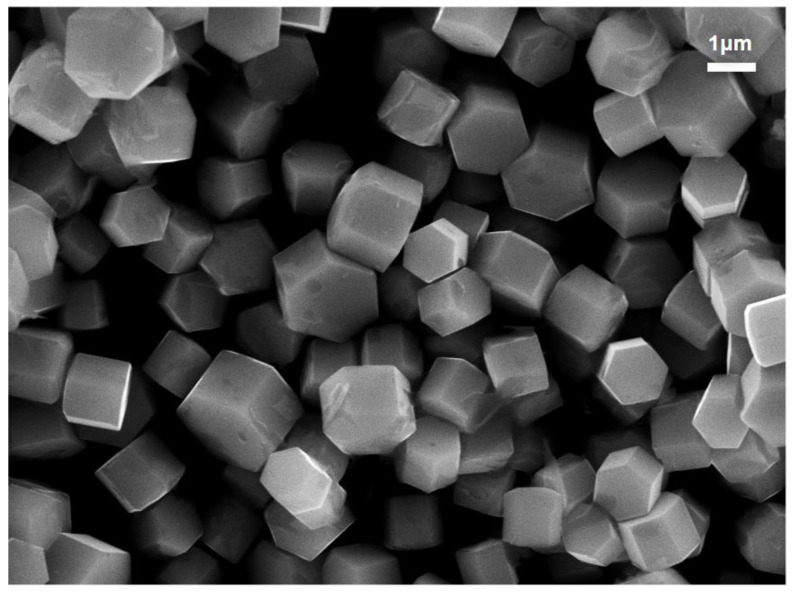
FE-SEM image depicting crystalline ZnO with characteristic hexagonal prism-like structures on the order of a few micrometers.

**Figure 2 nanomaterials-13-01652-f002:**
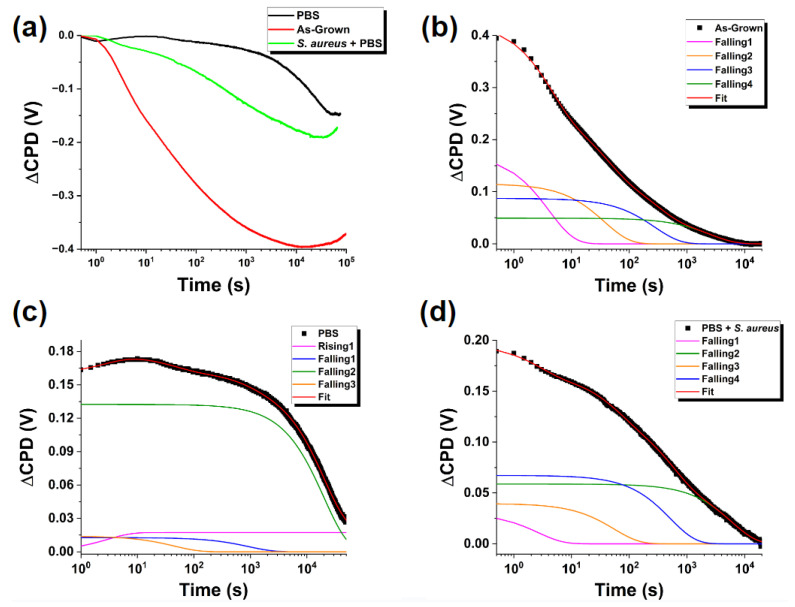
(**a**) SPV “light-on” transients under exposure to panchromatic light for ZnO MPs of balanced morphology, as-grown and exposed to indicated environments. (**b**–**d**) Fitting of experimental curves using Equation (1) for (**b**) as-grown ZnO MPs, (**c**) ZnO MPs after exposure to PBS solution, and (**d**) ZnO MPs after exposure to PBS solution containing *S. aureus* bacteria. The curves in (**b**–**d**) are shifted by the offset voltage *V*_0_.

**Figure 3 nanomaterials-13-01652-f003:**
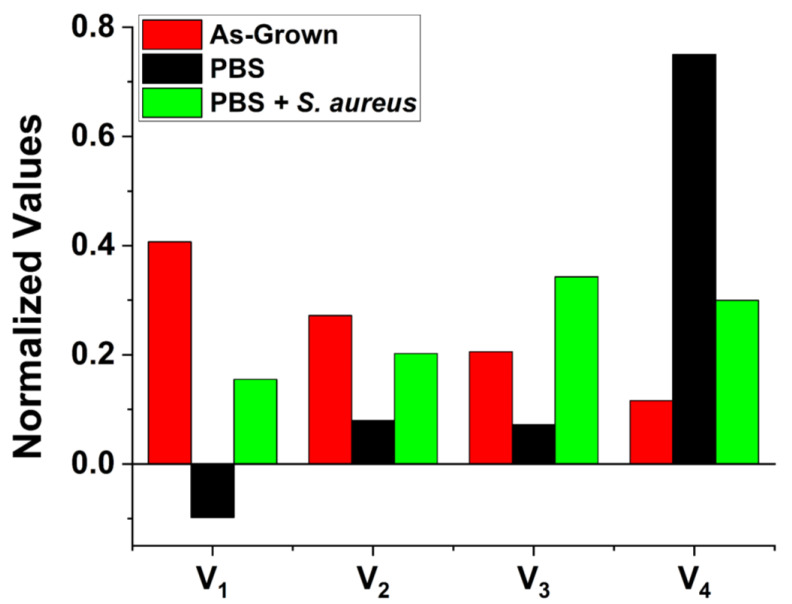
Normalized fitting parameters *V_i_* for as-grown ZnO MPs as well as those exposed to PBS with and without the presence of *S. aureus* bacteria. They were normalized to the sum of all *V_i_* for each sample. *V*_1_ is plotted negative for the sample exposed to PBS alone to indicate that this component has a directionality opposite to the others.

**Figure 4 nanomaterials-13-01652-f004:**
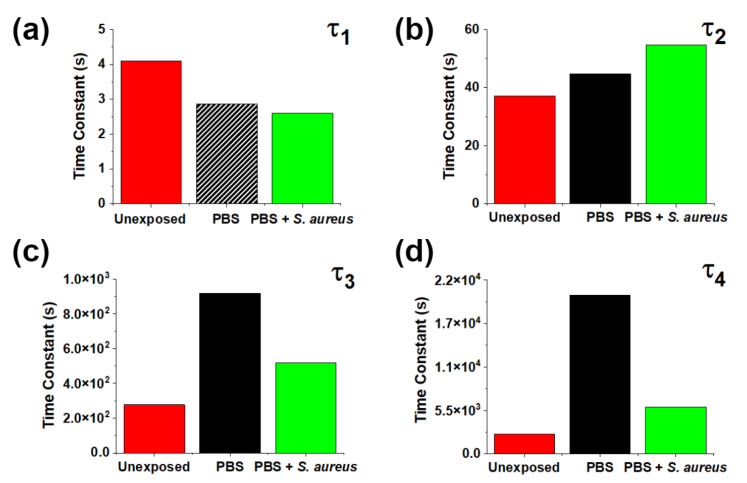
Fitting parameters (**a**) *τ*_1_, (**b**) *τ*_2_, (**c**) *τ*_3_, and (**d**) *τ*_4_ for characteristic SPV times for as-grown ZnO MPs and those exposed to PBS alone and with *S. aureus* bacteria. The hash pattern for *τ*_1_ indicates a “rising” process.

**Figure 5 nanomaterials-13-01652-f005:**
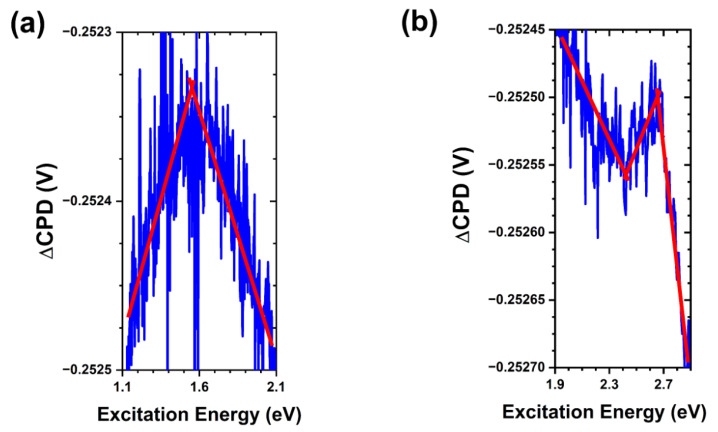
SPV spectral transitions in as-grown ZnO MPs of balanced morphology. (**a**) Trap-to-conduction band transition at ~1.5 eV. (**b**) Valence band-to-trap transition at ~2.4 eV and trap-to-conduction band transition at ~2.6 eV. Note that the red lines are a guide for the eye.

**Figure 6 nanomaterials-13-01652-f006:**
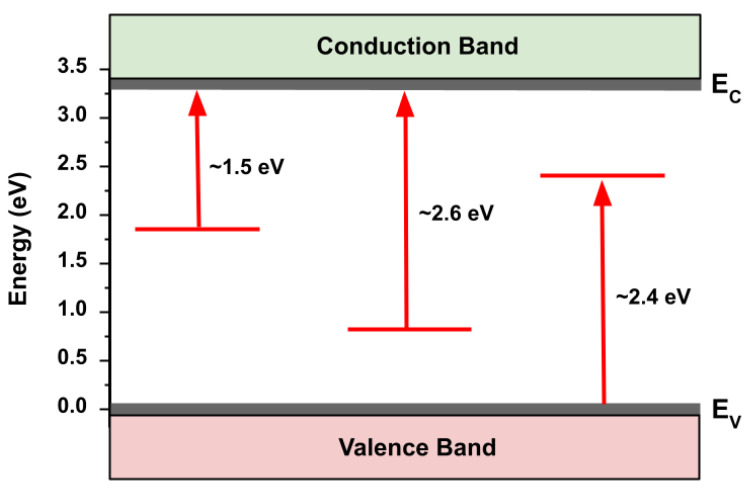
Energy diagram depicting sub-bandgap transitions observed in the SPV spectra of as-grown ZnO MPs of balanced morphology.

**Figure 7 nanomaterials-13-01652-f007:**
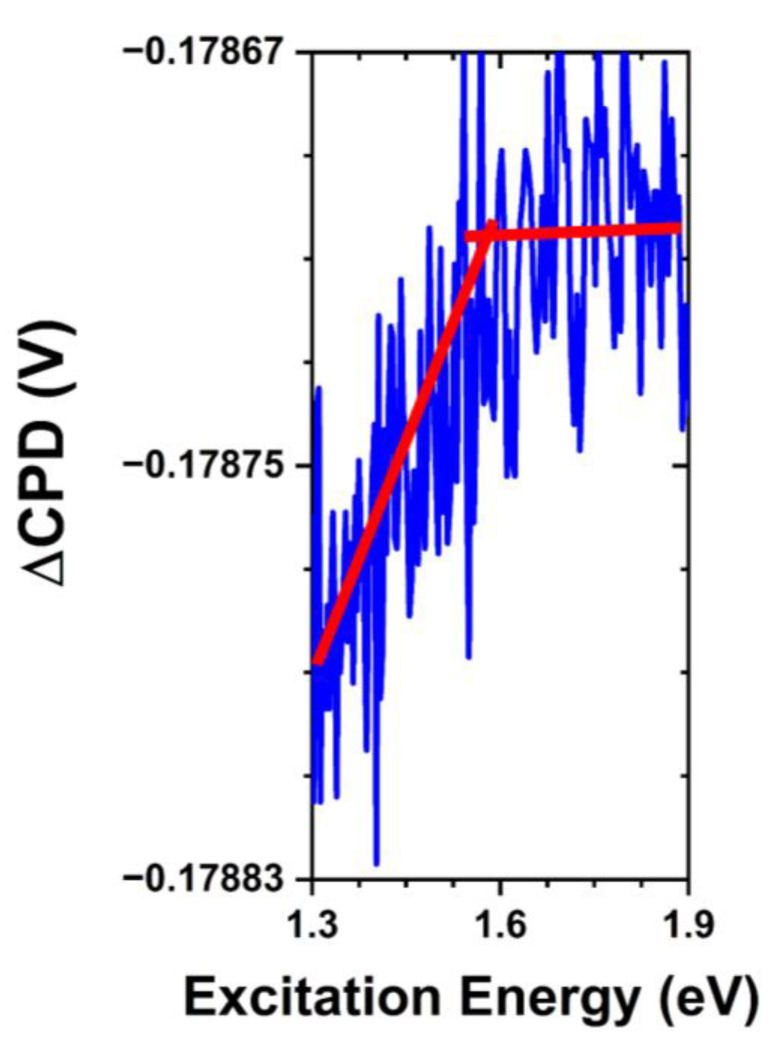
SPV spectral transition in hydrothermally grown ZnO MPs of balanced morphology after exposure to PBS without the presence of *S. aureus* bacteria, depicting a trap-to-conduction band transition at ~1.5 eV. Note that the red lines are a guide for the eye.

**Figure 8 nanomaterials-13-01652-f008:**
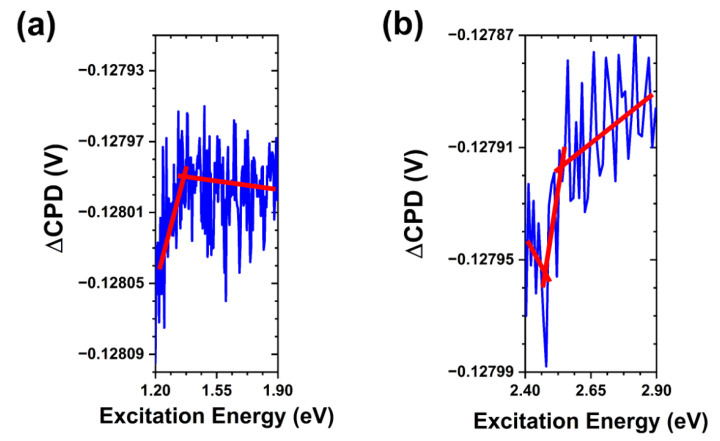
SPV spectral transitions in ZnO MPs of balanced morphology after exposure to *S. aureus* bacteria in PBS. (**a**) Trap-to-conduction band transition at ~1.4 eV. (**b**) Valence band-to-trap transition at ~2.45 eV and trap-to-conduction band transition at ~2.55 eV. Note that the red lines are a guide for the eye.

**Table 1 nanomaterials-13-01652-t001:** Fitting parameters for ZnO MPs as-grown and exposed to PBS with and without the presence of *S. aureus* bacteria. ^†^ Indicates parameters associated with processes described by rising exponentials.

	As-Grown	PBS	PBS + *S. aureus*
*V*_0_(V)	**−3.96 × 10^−1^**	**−1.75 × 10^−1^**	**−1.88 × 10^−1^**
	±3.04 × 10^−4^	±3.56 × 10^−4^	±2.42 × 10^−4^
*V*_1_(V)	**1.73 × 10^−1^**	**1.73 × 10^−2 †^**	**3.04 × 10^−2^**
	±1.03 × 10^−3^	±2.74 × 10^−4^	±2.12 × 10^−4^
*τ*_1_(s)	**4.09**	**2.87 ^†^**	**2.61**
	±2.95 × 10^−1^	±1.02 × 10^−1^	±0.04
*V*_2_(V)	**1.15 × 10^−1^**	**1.40 × 10^−2^**	**3.97 × 10^−2^**
	±1.32 × 10^−3^	±2.03 × 10^−4^	±3.96 × 10^−4^
*τ*_2_(s)	**3.71 × 10^1^**	**4.48 × 10^1^**	**5.47 × 10^1^**
	±9.16 × 10^−1^	±1.55	±1.05
*V*_3_(V)	**8.73 × 10^−2^**	**1.27 × 10^−2^**	**6.73 × 10^−2^**
	±1.32 × 10^−3^	±1.85 × 10^−4^	±4.11 × 10^−4^
*τ*_3_(s)	**2.77 × 10^2^**	**9.19 × 10^2^**	**5.19 × 10^2^**
	±9.13	±3.31 × 10^1^	±7.51
*V*_4_(V)	**4.93 × 10^−2^**	**1.32 × 10^−1^**	**5.89 × 10^−2^**
	±1.05 × 10^−3^	±1.87 × 10^−4^	±3.20 × 10^−4^
*τ*_4_(s)	**2.53 × 10^3^**	**2.01 × 10^4^**	**5.97 × 10^3^**
	±9.02 × 10^1^	±1.00 × 10^2^	±9.50 × 10^1^

## Data Availability

The data presented in this study are available on request from the corresponding author.

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
