# Peer review of "Surface Photovoltage Response of ZnO to Phosphate-Buffered Saline Solution with and without Presence of Staphylococcus aureus"

_nanomaterials, 2023, doi:10.3390/nano13101652_

Round 1

Reviewer 1 Report

The authors have done very good analysis. The device can be a potential tool for bacterial contamination test. 

Please consider following comments

Introduction,

There is a single paragraph, please break to four paragraphs.

Line 46 

The first paragraph should be end at line 46. please include some references at end of  sentence of ' Powerful bacterial adaptive responses in the form of genetic mutations and lateral gene transfer makes the development of effective traditional '.

please include references at end of  sentence of 'There exists a debate surrounding the most suggested mechanisms, namely: generation of various reactive oxygen species (ROS), release of Zn ions and surface-surface interactions between the bacterial cell wall and free crystalline surface of ZnO. The relative dominance of these mechanisms is also subject for  debate thus inhibiting efficient design of novel ZnO-based antimicrobials.'

The second paragraph should be made it at end of line 74

The third paragraph should be made at end of line 99. 

please include reference at end of sentences from line 77 to line 99 

please include results from literature what the range of vortag in literature with and without bacteria, which type of bacterial have been tried in the literature.

Please see the following paper

doi/full/10.1021/acsami.1c20198

 https://doi.org/10.1128/AEM.01015-2

method 

figure 4, please include scale bar for the column in the graphs

Author Response

Introduction,

There is a single paragraph, please break to four paragraphs.

We thank the reviewer for this suggestion. We agree that the introduction would be better presented in several paragraphs. We have therefore split it into four paragraphs as requested.

Line 46

The first paragraph should be end at line 46.

We appreciate the reviewer going through and suggesting potential locations to begin new paragraphs. We agree that this is a good location for doing so and have made the requested change.

Please include some references at end of  sentence of ‘ Powerful bacterial adaptive responses in the form of genetic mutations and lateral gene transfer makes the development of effective traditional ‘.

The reviewer makes a good suggestion to support the claims made in the quoted sentence. We have therefore added relevant references as requested.

please include references at end of  sentence of ‘There exists a debate surrounding the most suggested mechanisms, namely: generation of various reactive oxygen species (ROS), release of Zn ions and surface-surface interactions between the bacterial cell wall and free crystalline surface of ZnO. The relative dominance of these mechanisms is also subject for  debate thus inhibiting efficient design of novel ZnO-based antimicrobials.’

We appreciate the suggestion and have added references at the end of the quoted sentence.

The second paragraph should be made it at end of line 74

The third paragraph should be made at end of line 99.

Again, we appreciate the reviewer’s suggestion and introduced the requested paragraphs.

please include reference at end of sentences from line 77 to line 99

We thank the reviewer for this observation and have added relevant references to the suggested passage.

please include results from literature what the range of vortag in literature with and without bacteria, which type of bacterial have been tried in the literature.

We assume the reviewer is asking for literature results of typical voltage ranges. To our knowledge, no other studies are looking at the surface photovoltage (SPV) of metal oxides after exposure to bacteria or bacterial environments, let alone ZnO and S. aureus specifically. If the reviewer is aware of such a study, we would be happy to read it and compare our values.

Please see the following paper

doi/full/10.1021/acsami.1c20198

We thank the reviewer for suggesting we read this work. This paper presents an interesting and valuable method for improving antimicrobial air filters with low-voltage Laser-induced graphene.

 https://doi.org/10.1128/AEM.01015-2

While we appreciate the recommendation, this link leads to an error message from doi.org stating “DOI NOT FOUND”. If the reviewer feels that our reading of this paper is critical, then please produce a functional link.

figure 4, please include scale bar for the column in the graphs

We appreciate the reviewer’s remarks. The axes of all plots in the figure are accurately scaled and labeled. They are also presented in order of increasing timescale. Any clarification the reviewer may provide would be greatly appreciated.

Reviewer 2 Report

Dear Authors,

The paper presents interesting aspects related to Staphylococcus aureus.

Still several aspects should be solved:

1.       In the conclusion section the authors say: “Nano and microscale ZnO-based antibacterial agents represent a viable and attractive candidate for novel applications.”, but no examples of such applications are given. Please add this type of information.

2.       Also, please compare numerically the obtained results with the results obtained using other similar materials and approaches.

3.       The similarity of the paper is about 32% using Turnitin; a little to much to be published as such.
The beginning of the Introduction, Chapter 2.1 and 2.3 require attention.

I hope that my comments will be useful.

Sincerely,

Reviewer

Author Response

  1. In the conclusion section the authors say: “Nano and microscale ZnO-based antibacterial agents represent a viable and attractive candidate for novel applications.”, but no examples of such applications are given. Please add this type of information.

We thank the reviewer for this comment. We have added some applications with relevant references.

  1. Also, please compare numerically the obtained results with the results obtained using other similar materials and approaches.

We appreciate the reviewer’s comment. This is not done in this work because, to our knowledge, no other studies are looking at the surface photovoltage (SPV) of metal oxides after exposure to bacteria or bacterial environments, let alone ZnO and S. aureus specifically. If the reviewer is aware of such a study, we would be happy to read it and compare our values.

Despite this, there are areas where quantitative comparisons can be and are made. In the manuscript, we currently compare spectroscopic SPV results to those predicted by theoretical defect formation rates and energies predicted by Erhardt et. al.

  1. The similarity of the paper is about 32% using Turnitin; a little to much to be published as such.

The manuscript that we submitted is original research, and the results presented and discussed here were not reported in any other papers. If the reviewer can provide the sources that yield the 32% overlap in Turnitin, we will appreciate this.

The beginning of the Introduction, Chapter 2.1 and 2.3 require attention.

The beginning of the introduction is as follows:

“Wurtzite zinc oxide (ZnO) is an abundant type II-VI semiconductor with a wide, direct band gap of ~3.37 eV at room temperature, which corresponds to the spacing between the vacant 4s orbitals of Zn2+ and the occupied 2p states of O2- atoms within the crystalline lattice. Notable physical and chemical stability, in addition to a range of useful optoelectronic properties, has resulted in extensive and diverse applications in a variety of fields.”

It is a single sentence outlining the general relevant semiconductor characteristics of wurtzite ZnO followed by a brief sentence stating that it is a commonly used material with general support as to why. Neither statement is novel or groundbreaking nor are they intended to be so.

Sections 2.1 and 2.3 are methodological descriptions of standard experimental techniques with common instrumentation and materials, partially described by us in our previous publications.

Reviewer 3 Report

1) Focus on the English language.  For example: In this investigation, we utilize ZnO particles synthesized  (the comma was missing), ZnO-free surface ( It was a compound noun), etc.

2) The references are clustered. Explain each reference independently. 

3) Line 199: We used this model to fit the transient curves ( which  model are talking about ?)

 4)  Provide a notation table. 

1)  Spelling 

2) Punctuation 

3) Compound noun formation 

4) Synatax error 

5) Keywords 

Semi-colons are not used in this way as the authors used. 

Author Response

Comments and Suggestions for Authors

1) Focus on the English language.  For example: In this investigation, we utilize ZnO particles synthesized  (the comma was missing), ZnO-free surface ( It was a compound noun), etc.

We appreciate the reviewer’s input. We have added the requested comma. Many other similar errors were discovered upon additional review and have been corrected accordingly.

Regarding the compound noun remark, we have rewritten the sentence in question as follows: “…the free surface of ZnO…”. We make similar adjustments to this phrase for each appearance within the manuscript.

Numerous pronoun and article inconsistencies were found upon an additional review by a native English speaker. These significant issues have been corrected.

2) The references are clustered. Explain each reference independently.

Sorry, but we didn’t understand in which part of the manuscript the clustering occurred. Feedback from the reviewer would be appreciated.

3) Line 199: We used this model to fit the transient curves ( which  model are talking about ?)

We thank the reviewer and have changed the sentence to make it explicitly clear as follows:

“We used the model based on (Eq. 1) to fit the transient curves and thus quantify the observed changes.”

 4)  Provide a notation table.

While we appreciate the suggestion, the structure and allowed subsections are clearly outlined in section 3.1 of MDPI’s style guide as follows:

“3.1. Overall Structure

Research articles have a standard structure, which is set out in the instructions for authors of the journal and the journal template. The majority of journals use a so-called IMRAD structure, meaning that the sections are Introduction, Materials and Methods, Results, and Discussions. Some journals require a Conclusions section at the end, and others have the Materials and Methods section after the Results and Discussions. Authors may choose to have Results and Discussions as one or two sections.”

We are abiding by the journal’s formatting requirements regarding acronyms and abbreviations. These can be found on the journal website and are copied below:

“Acronyms/Abbreviations/Initialisms should be defined the first time they appear in each of three sections: the abstract; the main text; the first figure or table. When defined for the first time, the acronym/abbreviation/initialism should be added in parentheses after the written-out form.”

Comments on the Quality of English Language

1)  Spelling

A check by native English speakers in addition to software verification returned no spelling errors. If the reviewer has a specific error in mind, please let us know so we may make this correction.

2) Punctuation

We thank the reviewer for this suggestion. Numerous commas have been added throughout the text to improve the manuscript's grammatical correctness and overall clarity.

3) Compound noun formation

Upon additional review by a native English speaker, incorrect compound noun syntax was utilized throughout. This issue has been addressed. Below is a list of the relevant phrases:

“phosphate-buffered saline”

“surface-specific responses”

“longer-lived processes”

“oxygen-rich sites”

“well-fitted”

4) Synatax error

Such errors have been remedied.

5) Keywords

Keywords are listed and separated by semicolons according to the Nanomaterials template and following MDPI style guidelines.

Semi-colons are not used in this way as the authors used.

The only usage of semicolons is in the listing of keywords, authors in the affiliations, and contributions as per the Nanomaterials template following MDPI style guidelines. These are copied below from the MDPI style guide, section 5.4:

“5.4. Colons and Semicolons

As mentioned above, em dashes are preferred to colons for introducing definitions. Colons may be used to introduce lists or before equations, but not where they separate a verb and its object or a preposition and its object.

Semicolons may be used in lists, as mentioned above. For other uses of semicolons, refer to a grammar book. In general, we recommend using semicolons sparingly and considering whether a period or comma would be more appropriate.”

Round 2

Reviewer 1 Report

The manuscript has been improved. Please change to two decimal of all of the numbers for the table 1.

What is delta CPV in the  Y axle in figure 5,7,8. Please describe it before the figure appear.

In general, there are figure legends and table 1, but there is no description  and  discussion of the data of figure 5,6,7,8.

Please describe the table 1 in the text.

Author Response

Please change to two decimal of all of the numbers for the table 1.

We thank the reviewer for the remarks. We have updated all values within the table to retain only 2 decimal points.

What is delta CPV in the  Y axle in figure 5,7,8. Please describe it before the figure appear.

The y-axes of these figures are the change in contact potential difference as defined on line 155. Originally the abbreviation read (VCPD) in the text. We thank the reviewer for noticing this inaccuracy in the text and have corrected the notation to line up with the figures.

In general, there are figure legends and table 1, but there is no description  and  discussion of the data of figure 5,6,7,8.

All the figures are described and discussed as follows.

Figure 5:

Discussed on lines 461 – 471: “The results of SPV spectroscopy for the as-grown sample are shown in Figure 5 and summarized in Figure 6. Here we demonstrate changes in the contact potential difference (ΔCPD) with respect to incident wavelength rather than time to elucidate the surface and near-surface electronic states of our material. In addition to a strong bandgap transition at ~3.3 eV, we observed the presence of several states in the bandgap: a ~1.5 eV trap-to-conduction band transition, a ~2.6 eV trap-to-conduction band transition, and a ~2.4 eV valence band-to-trap transition. Following theoretical calculations [38], we assign these transitions to oxygen vacancies, zinc vacancies, and oxygen interstitials respectively. Assignment of these surface trap states is important as it allows us to define the initial nature of the free crystalline surface upon which interactions with bacteria and media components may take place.”

Captioned on Lines 473 – 475: “Figure 5. SPV spectral transitions in as-grown ZnO MPs of balanced morphology after exposure to S. aureus bacteria in PBS. (a) Trap-to-conduction band transition at ~1.5 eV. (b) Valence band-to-trap transition at ~2.4 eV and trap-to-conduction band transition at ~2.6 eV.”

Figure 6:

Figure 6 is a bandgap representation of Figure 5 and is therefore discussed alongside Figure 5.

Figure 7:

Discussed on lines 479 – 492: “SPV spectra collected following the exposure to PBS with and without the presence of S. aureus bacteria demonstrated the effects on the surface electronic structure of ZnO. In addition to the anticipated weakening of the bandgap transition due to the partial dissolution of the surface and adsorption of media components, we find notable differences in the observed surface states, which we present in Figure 7. Transitions associated with the oxygen-rich states are suppressed following PBS exposure while the ~1.5 eV transition associated with the oxygen-deficient states is preserved. There have been several investigations into the kinetics of phosphate adsorption, and it is widely attributed to the ligand exchange with surface hydroxyl groups resulting in PO43- adsorption [23,39]. Considering the highly reactive nature of an oxygen-rich surface in aqueous environments, it is clear that the formation of such hydroxyls and subsequent ligand exchange is a probable explanation for the removal of these states. Therefore, phosphate adsorption is likely to take place at these previously oxygenated sites thus amplifying the slow charge-exchange processes and reversing the direction of the fast charge transfer discussed above.”

Continued on lines 497 – 503: “Considering the results of Lv et al. in tracking Zn2+ ion release as a measure of ZnO particle solubility in aqueous phosphate environments [23], one can surmise that the high concentration of phosphates in a solution likely stabilizes the crystalline surface through the adsorption processes in addition to reacting with the initial rapid release of Zn2+ ions. This would have significant effects on the antibacterial action of these particles and supports previous studies linking phosphate-rich media with decreased bacterial growth inhibition [21,22,40,41].”

Captioned on Lines 494 – 496: “Figure 7. SPV spectral transition in hydrothermally grown ZnO MPs of balanced morphology after exposure to PBS without the presence of S. aureus bacteria depicting a trap-to-conduction band transition at ~1.5 eV.”

Figure 8:

Discussed on lines 461 – 471: “Comparing these results with the SPV spectra obtained following the exposure to PBS containing S. aureus bacteria we again observe a similar weakening of the bandgap transition due to the partial dissolution of ZnO crystals and adsorption of media components. However, we do not see significant changes to the sub-bandgap transitions observed in the sample exposed to PBS alone. In Figure 8 we see all three transitions that were originally present in the as-grown ZnO MPs. This lines up with the observed effects of bacteria containing PBS on the SPV transients discussed above where we found little indication of new or suppressed states. The point of interest in these measurements lies in the apparent inhibition of the effects of PBS alone outlined thus far. The preservation of the oxygen-rich states is evidence of bacteria limiting phosphate adsorption at the free crystalline surface. This indicates that either bacteria significantly reduce the concentration of aqueous phosphates in the local environment or more likely that they promote increased solubility of the ZnO crystals through interactions at the crystalline surface thus shifting the dominant phosphate interactions to reaction with Zn2+ in solution as opposed to surface adsorption. Such surface interactions are supported by the previously reported surface degradation and changes in the photoluminescence intensity after exposure to antibacterial interactions [22]. These findings suggest that bacteria promote Zn2+ ion release and this release is limited by aqueous phosphates whereas interaction with the released ions is competitive in nature between bacteria and environmental phosphates. Bacteria and phosphates competing for interaction at the surface sites is unlikely due to the difference in their interactions at these states. Restriction of these sites by adsorbed species could limit surface reactive oxygen species (ROS) generation. Nevertheless, one would anticipate observation of a state that is suppressed by PBS but partially restored in the presence of S. aureus bacteria under the commonly proposed UV-mediated photocatalysis route of ROS generation in ZnO.”

Captioned on Lines 473 – 475: “Figure 5. SPV spectral transitions in as-grown ZnO MPs of balanced morphology after exposure to S. aureus bacteria in PBS. (a) Trap-to-conduction band transition at ~1.5 eV. (b) Valence band-to-trap transition at ~2.4 eV and trap-to-conduction band transition at ~2.6 eV.”

Please describe the table 1 in the text.

Discussed on lines 210 – 221: We have compiled the results of these fits in Table 1 and Figure 2 (b)-(d). It is worth noting that all the charge exchange processes were well fit with four decaying exponentials except for the “fastest” process in the sample exposed to PBS alone, which is fitted with a rising exponential. In comparing the parameters across the samples, it becomes apparent that the PBS medium significantly impacts the surface charge dynamics on all the time scales. The presence of S. aureus bacteria qualitatively alters the results for both Vi and τi in comparison to the case of exposure to PBS alone.

When considering the physical origins of these exposure-induced changes in the charge dynamics, there is a possibility of introducing new surface states and modifying existing ones. Especially prominent are the changes seen in the normalized V4 parameter (see Figure 3). Generally, in our fitting model, the Vi parameters represent the effective charge “reservoir” of these processes analogous to the charge of a capacitive element.”

Reviewer 3 Report

Check  line  41 [2-6]

I think it is not a necessary condition to have a long reference list. Even if the authors provide 20 references, it would be sufficient, but it must be relevant to the study. That was my main concern. I do not think that the references [2], [3], [4], [5], and [6] pitched the same idea. 

I do not agree with the definition of semi-colon. 

Semi-colon is used to separate pairs of words and the keywords are not pairs of words. 

It is not even an independent clause that is related to another sentence. 

Author Response

Comments and Suggestions for Authors

Check  line  41 [2-6]

I think it is not a necessary condition to have a long reference list. Even if the authors provide 20 references, it would be sufficient, but it must be relevant to the study. That was my main concern. I do not think that the references [2], [3], [4], [5], and [6] pitched the same idea.

We thank the reviewer for this suggestion. We have reduced the number of references with overlapping content as requested.

Comments on the Quality of English Language

I do not agree with the definition of semi-colon.

Semi-colon is used to separate pairs of words and the keywords are not pairs of words.

It is not even an independent clause that is related to another sentence.

This reviewer’s concerns should be addressed to the Editorial Board, which defines the Journal’s style choices. Articles published in this particular journal employ semi-colons to separate keywords.